# Granulocytic Myeloid-Derived Suppressor Cells in Breast Milk (BM-MDSC) Correlate with Gestational Age and Postnatal Age and Are Influenced by Infant’s Sex

**DOI:** 10.3390/nu12092571

**Published:** 2020-08-25

**Authors:** Natascha Köstlin-Gille, Lara-Antonia Flaig, Marco Ginzel, Jörg Arand, Christian F. Poets, Christian Gille

**Affiliations:** Department of Neonatology, Tuebingen University Children’s Hospital, 72076 Tuebingen, Germany; lara-antonia.flaig@student.uni-tuebingen.de (L.-A.F.); marco.ginzel@med.uni-tuebingen.de (M.G.); joerg.arand@med.uni-tuebingen.de (J.A.); christian-f.poets@med.uni-tuebingen.de (C.F.P.); christian.gille@med.uni-tuebingen.de (C.G.)

**Keywords:** MDSC, breast milk, breastfeeding, preterm infants, neonate

## Abstract

Background: Infections are the main cause of death in preterm infants. Causative agents often descend from the intestinal flora of the infected neonate, indicating insufficient protection by the mucosal barrier. Breast milk (BM) contains different subsets of immune cells. We recently showed that BM contains significant numbers of myeloid-derived suppressor cells (MDSC)—immune cells that actively suppress pro-inflammatory immune responses—and hypothesized that the transfer of BM-MDSC may modulate the mucosal immunity of the newborn. Methods: Percentages of MDSC in the BM from mothers of 86 preterm infants between 23 + 0 and 36 + 6 weeks of gestation during their first five postnatal weeks were analyzed by flow cytometry and correlated with maternal and infant characteristics. Results: Percentages of BM-MDSC positively correlated with gestational age and postnatal age. The expression of activation markers on BM-MDSC did not change with gestational age, but it decreased with postnatal age. Mothers who received antepartum tocolytics had lower percentages of BM-MDSC, and infant’s sex strongly influenced percentages of BM-MDSC. Conclusion: Our results point toward a role of BM-MDSC for immune regulation in the neonatal gut, making them a potential target of immune-based therapies shortly after birth.

## 1. Introduction

Infections are one of the most important complications in the care of preterm infants and often lead to death or long-term sequelae. About one out of three preterm infants suffers from at least one episode of infection during his stay in the neonatal intensive care unit (NICU) [1]. For both neonatal sepsis and necrotizing enterocolitis (NEC), which are the most important infections in preterm infants, it has been shown that alterations in the intestinal microbiome precede disease onset and that causative pathogens often descend from the intestinal flora of the infected infant itself [2,3,4,5], highlighting specific defects in mucosal immunity. 

Breast milk (BM) is thought to be the optimal nutrition for neonates. Besides all essential nutritional components, BM contains non-nutrient immunological factors such as immunoglobulins, antimicrobial proteins, and immune cells that may promote microbiome establishment and protect against acute respiratory, gastrointestinal, and systemic infections [6,7,8,9,10]. In addition, breastfeeding is also associated with a reduced risk of chronic inflammatory diseases in later life such as asthma, atopy, diabetes, obesity, and inflammatory bowel disease [11,12,13,14,15], suggesting long-term immunomodulatory effects. Mechanisms regulating these effects are incompletely understood. Corresponding to the nutrient composition of BM, which adapts to the needs of the neonate, the content of immune cells and immunological factors in BM may vary over time as well. For example, it could have been shown that maternal and infants’ infections lead to a change in immune cell composition of BM [16] pointing toward a double-sided influence of both mothers and infants’ characteristics on BM immunity.

Myeloid-derived suppressor cells (MDSC) are myeloid cells with suppressive activity on other immune cells, especially on T-cells [17]. Depending on their phenotype, they can be sub-grouped into two populations: monocytic MDSC (MO-MDSC) and granulocytic MDSC (GR-MDSC). Primarily, the accumulation of MDSC has been described in cancer patients and tumor-bearing mice where they inhibit anti-tumor T-cell responses, leading to disease progression [18,19]. Later, it was shown that MDSC and especially GR-MDSC also accumulate physiologically during pregnancy in the fetal and maternal organism [20,21,22,23,24,25], mediating materno-fetal tolerance. Recently, we could show that also BM contains a large population of GR-MDSC (BM-MDSC) [26], leading to the hypothesis that the transfer of BM-MDSC from mother to baby may influence the mucosal immunity of the newborn. However, to date, nothing is known about the content of GR-MDSC in BM of preterm infants. Furthermore, nothing is known about how antenatal maternal treatment and infant characteristics influence the presence of GR-MDSC in BM.

In the present study, we quantified GR-MDSC in the breast milk of preterm infants between 23 + 0 and 36 + 6 weeks of gestation (WOG) during their first five postnatal weeks and correlated them with gestational age and postnatal age. Furthermore, we analyzed the expression of activation markers on BM-MDSC and analyzed how maternal and fetal characteristics (birth mode, antepartum steroids, antepartum tocolysis, multiparity, and infants’ sex) influenced BM-MDSC expression.

## 2. Materials and Methods

### 2.1. Patients

The local ethics committee approved this study, and all parents gave written informed consent (protocol number 682/2016BO1 and 419/2019BO2). From February 2018 to May 2020, breast milk from the mothers of preterm infants (born between 23 + 0 WOG and 36 + 6 WOG, *n* = 105) was collected at the Department of Neonatology at Tuebingen University Hospital. Mothers with severe complications (infections, rheumatic diseases, malignancies) were excluded from the study. Breast milk samples (20–100 mL) were collected during the first five postnatal weeks once or twice a week. The first sample was taken at the time when the mother had enough milk to fully feed her baby. Only mothers who provided at least three milk samples were included in the study (*n* = 86).

### 2.2. Definitions

*Gestational age.* Gestational age was calculated based on early prenatal ultrasound and obstetric examination.

### 2.3. Cell Isolation and Flow Cytometry

Milk samples were stored at 4 °C until further use and processed within six hours after collection. The isolation of milk cells was done as described previously [26]. Breast milk was diluted in a 1:1 ratio with phosphate-buffered saline (PBS, Biochrom, Berlin, Germany) and centrifuged for 20 min at 805× *g*. The supernatant was removed, and cells were washed two times with PBS. Cell count was determined, and cells were diluted in PBS at a concentration of 1 × 10^6^ cells/mL for extracellular staining. GR-MDSC were characterized as CD66b^+^/CD33^+^/CD14^−^/HLA-DR^low/−^ cells, according to previously established human MDSC characterization methods [21,23]. The gating strategy is depicted in Appendix A. Antibodies used for extracellular staining were anti-CD66b-FITC (clone G10F, concentration 1 µL/1 × 10^5^ cells), anti-CD33-PE (clone WM53, concentration 1 µL/1 × 10^5^ cells), anti-HLA-DR-PerCP-Cy5.5 (clone REA805, concentration 0.1 µL/1 × 10^5^ cells), and CD14-APC (clone MφP9, concentration 1 µL/1 × 10^5^ cells) (purchased from BD biosciences, Heidelberg, Germany (CD66b, CD33 and CD14) and Miltenyi Biotec, Bergisch-Gladbach, Germany (HLA-DR)). Antibodies were tested for their specificity by isotype control staining when introduced in our laboratory. The viability stain of single samples after density gradient centrifugation revealed about 10% dead cells. Data acquisition was performed with a FACS Calibur flow cytometer (BD biosciences), and data were analyzed via FlowJo ^TM^ (Ashland, OR, USA) 10 software (BD biosciences).

### 2.4. Statistical Analysis

Statistical analysis was done using GraphPad Prism 8.0 (GraphPad Software, La Jolla, CA, USA). Data were analyzed for Gaussian distribution by D’Agostino–Pearson omnibus normality test. Since data were not normally distributed, differences between two groups were evaluated using the Mann–Whitney test. Differences between more than two groups were evaluated by the Kruskal–Wallis test and Dunn’s multiple comparisons test. Correlations were analyzed by Spearman correlation. For the correlation analysis of postnatal age with GR-MDSC, repeated measurements of the same infant were included. A *p*-value < 0.05 was considered statistically significant.

## 3. Results

### 3.1. Study Cohort

During the observational period, 86 mothers from 105 preterm infants were included in this study; 34 mothers had delivered at <28 0/7 WOG, 34 had delivered at 28 0/7–31 6/7 WOG, and 18 had delivered at >32 0/7 WOG. Table 1 and Table 2 show the maternal and infant characteristics of the study cohort.

### 3.2. Levels of GR-MDSC in Breast Milk Are Positively Correlated with Gestational Age

We first analyzed percentages of GR-MDSC in the breast milk of preterm infants in relation to gestational age (GA). Therefore, we collected breast milk samples weekly during the first five postnatal weeks and calculated mean percentages of GR-MDSC of all CD45^+^. Then, mean percentages of GR-MDSC were correlated with gestational age at birth. Appendix A shows the gating strategy for GR-MDSC in breast milk. We found that mean percentages of GR-MDSC positively correlated with gestational age at birth (*p* < 0.005, *r* = 0.3, *n* = 86, Figure 1A). Absolute cell counts (Appendix A) and percentages of CD45^+^ leucocytes (Appendix A) in breast milk did not correlate with gestational age. Then, mothers were divided into the above three gestational age groups. We found that percentages of GR-MDSC were lowest in group 1 (median 39.7%, *n* = 34) and increased in groups 2 and 3 (median 47.9%, *n* = 34, not significant, and 57.5%, *n* = 18, *p* < 0.05, both compared to group 1, Figure 1B). As infants born at <28 0/7 WOG are at the highest risk for complications during the neonatal period, we further analyzed the GR-MDSC content in the breast milk of these infants, which was this time further divided into two subgroups (<26 0/7 WOG and 26 0/7 WOG–27 6/7 WOG). As expected, mothers delivering at <26 0/7 WOG had again significantly lower percentages of GR-MDSC in their milk (median 32.5%, *n* = 18) than those delivering at 26 0/ WOG–27 6/7 WOG (median 50.1%, *n* = 16, *p* < 0.05, Figure 1C). The expression of previously described activation markers (CXC-motif chemokine receptor 4(CXCR4), programmed death ligand-1 (PD-L1), and programmed death ligand-2 (PD-L2)) of BM-MDSC [26] did not correlate with gestational age (Figure 1D–F).

### 3.3. Levels of GR-MDSC in Breast Milk Increase with Postnatal Age 

Next, we analyzed percentages of GR-MDSC in BM in relation to postnatal age. We received breast milk samples from 39 mothers (45%) in the first postnatal week, from 74 (86%) mothers in the second postnatal week, from 74 (86%) mothers in the third postnatal week, from 66 (77%) mothers in the fourth postnatal week, and from 35 (41%) mothers in the fifth postnatal week. Correlation analysis showed that postnatal age correlated slightly but significantly with percentages of BM-MDSC (*p* < 0.05, *r* = 0.14, *n* = 288) (Figure 2A). Separate analysis by postnatal week showed an increase in BM-MDSC between week 2 (median 35.0%, *n* = 74) and week 3 (median 49.0%, *n* = 74, *p* < 0.05), while levels of leucocytes in breast milk remained constant over the first 5 weeks of lactation (Appendix A). Comparison of the expression of activation markers CXCR4, PD-L1, and PD-L2 on BM-MDSC between week 1 and week 5 showed a decreased expression of CXCR4 (Figure 2C, median mean fluorescence intensity (MFI) 39.0 in week 1 and 24.0 in week 5, *n* = 36–45, *p* < 0.005) and PD-L2 (Figure 2E, median MFI 39.8 in week 1 and 21.0 in week 5, *n* = 36–45, *p* < 0.05) in week 5, while the expression of PD-L1 (Figure 2D) remained unchanged.

### 3.4. Decreased Levels of GR-MDSC in Breast Milk from Mothers Receiving Tocolytics

To evaluate whether GR-MDSC in BM are influenced by maternal factors, we analyzed percentages of BM-MDSC in mothers after spontaneous delivery versus caesarean section and in those that received antepartum steroids or tocolytics versus those who did not. We found that levels of GR-MDSC in breast milk tended to be higher in women who delivered by caesarean section (median 49.5% versus 36.7%, *n* = 11–38, Figure 3A). In contrast, mothers who received antepartum steroids tended to have lower levels of BM-MDSC than those who did not (median 42.1% versus 54.6%, *n* = 16–65, Figure 3B). However, these results did not reach statistical significance. In addition, women who received tocolytics had significantly lower BM-MDSC levels than those who did not (median 40.2% versus 51.8%, *n* = 34–39, *p* < 0.05, Figure 3C). 

### 3.5. GR-MDSC Levels in Breast Milk Depend on Baby’s Gender

Lastly, we looked into infant characteristics potentially influencing BM-MDSC levels. Surprisingly, we found significantly higher MDSC-levels in BM given to daughters than sons (median 55.9% versus 42.5%, *n* = 47–59, *p* < 0.005, Figure 4A). Multiparity had no influence on percentages of BM-MDSC (multiples median 54.4% versus singletons median 47.4%, *n* = 18–68, Figure 4B).

## 4. Discussion

The immune modulatory role of breast milk and the transfer of immune cells by breast milk have been known for many years [16,27]. Recently, our group identified GR-MDSC in the BM of term and preterm infants [26], giving rise to the hypothesis that BM-MDSC may play a role in intestinal neonatal immune regulation. However, until now, nothing was known about the dynamics of GR-MDSC in the BM from mothers of preterm infants. Thus, we set out to investigate the expression of BM-MDSC in the BM of preterm infants depending on maternal and infant clinical characteristics. We found that (1) percentages of GR-MDSC in the BM of preterm infants were high overall and correlated positively with gestational and postnatal age (albeit less so in the latter), while the expression of activation markers on BM-MDSC did not; and (2) the expression of activation markers decreased with postnatal age. (3) The BM of mothers who received antepartum tocolytics contained lower levels of BM-MDSC than that of mothers who did not, while the mode of delivery and administration of antepartum corticosteroids had no influence on percentages of BM-MDSC. Lastly, we found that (4) BM fed to girls contained significantly more BM-MDSC than that given to boys, while multiple births did not influence BM-MDSC expression. 

Our finding that percentages of BM-MDSC increased with gestational age is in contrast to results from previous studies, where we found no correlation between GR-MDSC levels in the cord blood of preterm infants [24] or the peripheral blood of pregnant women [21] with gestational age. To our knowledge, only one study investigated leucocyte numbers in relation to gestational age and found a positive correlation of granulocyte counts with gestational in transitory milk (week 2) but not in mature milk (after week 2) [28]. At this point, an important limitation of our study has to be mentioned: because of the lack of specific surface markers, no phenotypic discrimination between GR-MDSC and mature neutrophils in BM was possible. We used the granulocytic marker CD66b for the characterization of BM-MDSC; however, CD66b is also expressed on neutrophils. In peripheral blood, GR-MDSC differ from mature neutrophils by sedimentation in the low-density fraction upon centrifugation, but due to high cell loss, density gradient centrifugation was not possible in our study. The main characteristic of GR-MDSC that distinguishes them from neutrophils is their immunosuppressive capacity. In our previous study [26] and with single samples from the present study (data not shown), we found that CD66b^+^-enriched milk cells possessed even stronger suppressive activity than GR-MDSC from the peripheral blood, suggesting that most of these cells are not granulocytes but indeed GR-MDSC. Thus, we regarded CD66b^+^ cells in BM as BM-MDSC.

Especially low percentages of GR-MDSC were found in BM of preterm infants born before 26 + 0 weeks of gestation, which is the cohort of patients with the highest risk for infections and NEC. However, due to the relatively small study population and the diversity in treatment (e.g., antibiotic therapy, probiotics) and nutrition (e.g., short-term pasteurization of BM to avoid the transmission of cytomegalovirus) of the infants, it was not possible to correlate percentages of BM-MDSC with neonatal complications. Further studies are necessary to analyze the impact of BM-MDSC on microbiome establishment, neonatal inflammatory diseases, and chronic inflammatory diseases in later life. Analysis of the expression of activation markers on BM-MDSC showed no change in PD-L1, PD-L2, and CXCR4 expression in relation to gestational age, suggesting that increased percentages of BM-MDSC do not come along with decreased activity.

Our finding of a positive association between age and BM-MDSC is also in contrast to previous studies from our group showing that GR-MDSC levels in pregnant women fall immediately after birth to levels found in non-pregnant controls [21]. In neonates, we also observed increased percentages of GR-MDSC only until 28 days of postnatal life [24]. More in line with our current findings, Trend et al. also found increased granulocyte counts in BM between colostrum and mature milk [28]. Thus, we assume that systemic GR-MDSC as a relic from in utero life may be detrimental by contributing to the increased susceptibility to infections in newborns, while BM-MDSC are continuously transferred to the infant and may be needed for the local modulation of immune activation in the gut, where tolerance toward the establishing microbiome is necessary. The expression of CXCR4 and PD-L2 on BM-MDSC decreased with rising postnatal age, pointing toward a decrease in their suppressive activity. However, we previously found a very strong inhibitory capacity of BM-MDSC isolated six weeks after parturition [26], illustrating that they are still very effective.

As inflammation has been implicated in the initiation of preterm and term labor [29] and inflammation is one of the most important signals for the expansion of immature myeloid cells which later can be converted to MDSC [30], we analyzed GR-MDSC in BM in relation to the mode of delivery and in relation to the administration of antepartum tocolytics and found that percentages of GR-MDSC were decreased in the BM of mothers who received antepartum tocolytics and tended to be lower than that of mothers who delivered spontaneously. One explanation for this finding could be that the expression of pro-inflammatory cytokines such as Interleukin-8 or Interleukin-6 during labor may lead to so-called “emergency myelopoesis”, emptying the storage pools for myeloid progenitors in the bone marrow [31] and thereby diminishing the capacity to later produce MDSC for BM. Conversely, preterm labor is accompanied by a decline in the sex hormone progesterone, which has been described to positively regulate MDSC homeostasis [32,33]. Thus, decreased GR-MDSC levels in the BM of women with preterm labor could also result from decreased progesterone levels in these individuals. In our cohort, the first-line drug for intravenous treatment of preterm labor was magnesium–sulfate. One small study showed that the application of magnesium sulfate to the mother reduced the chemotaxis and motility of neutrophils in preterm infants [34]. However, functional studies investigating the direct effects of tocolytics such as magnesium sulfate on MDSC are lacking. It remains unclear whether the observed decrease of GR-MDSC in BM of mothers who received antepartum tocolytics is due to direct effects of the medication or indirect effects of factors associated with preterm labor.

Glucocorticoids are known to have anti-inflammatory effects and are routinely used to prevent respiratory distress syndrome (RDS) in neonates when preterm delivery is expected. Therefore, we analyzed the effect of administration of antenatal steroids on percentages of GR-MDSC in BM. We found that antenatal steroids had no significant effect on the expression of BM-MDSC. Several studies showed that betamethasone (the most common used drug for RDS prophylaxis) had profound effects on neonatal immune cells in the sense of immune suppression [35,36,37,38]. Furthermore, it is known that glucocorticosteroids lead to neutrophilia by inducing neutrophil maturation in and mobilization from the bone marrow but decrease the ability of neutrophils to migrate to inflamed tissue [39]. Thus, the recruitment of neutrophilic cells such as GR-MDSC from the periphery to the breast milk may be hampered as well, explaining the tendency toward lower GR-MDSC levels in the BM of women who received antepartum steroids. At this point, another limitation of our study has to be mentioned; due to the descriptive design, it only can be speculated how differences in percentages of BM-MDSC between groups could be explained. Mechanistic studies are needed to further investigate the impact of maternal treatment on BM-MDSC accumulation and function.

Infant’s sex is well known as an important prognostic factor in preterm infants [40,41]. Preterm boys are more vulnerable to postnatal complications such as respiratory distress syndrome and poor neurodevelopmental outcome than girls [42]. While the main cause for this bias is unclear, it has been speculated that early life nutritional stimuli may play a role [42]. Various studies showed that the nutritional composition of BM differs between the mothers of male and female infants with higher fat content in the BM of females and higher salt content in the BM of males [43] as well as different concentrations in human milk oligosaccharides [44] and free amino acids [45] between males and females. However, to our knowledge, until now, no studies exist that investigate the content of immune cells or immune mediators in BM in relation to infants’ gender. In the present study, we found that mothers of daughters had significantly higher levels of BM-MDSC than mothers of sons. Until now, the functional role of BM-MDSC has been poorly understood. Recently, we showed that in vitro BM-MDSC are able to inhibit T-cell proliferation and TLR4 expression by monocytes [26]; however, the long-term effects of GR-MDSC in BM are unclear. Assuming a beneficial role of BM-MDSC for neonatal immunity, the increased GR-MSC levels in the BM of female infants may contribute to their better clinical outcome. The mechanism of how the backward stimulation of leucocyte recruitment to BM is mediated is unclear. Hassiotou et al. showed that infant’s infections can stimulate a leucocyte response in BM [16]. They speculated that a backwards milkflow during suckling may transfer infants’ saliva together with microorganisms to the breast, thereby stimulating a local immune response [16]. Corresponding to this, small amounts of female sex hormones could be transferred that may recruit GR-MDSC. As mentioned before, the BM of female infants contains higher amounts of fat than that of males [43]. Unpublished data from our group reveal that fatty acids can activate GR-MDSC. Thus, the activation of GR-MDSC by BM fat may another potential mechanism for increased GR-MDSC levels in the BM of female infants.

Taken together, we show that GR-MDSC levels in BM are not stable but vary depending on gestational and postnatal age, maternal antepartum therapy, and infant sex. Further studies are needed to figure out the role of BM-MDSC for immune regulation during the neonatal period. Targeting BM-MDSC especially in preterm infants at high risk for inflammatory diseases may be a strategy to improve neonatal outcome.

## Figures and Tables

**Figure 1 nutrients-12-02571-f001:**
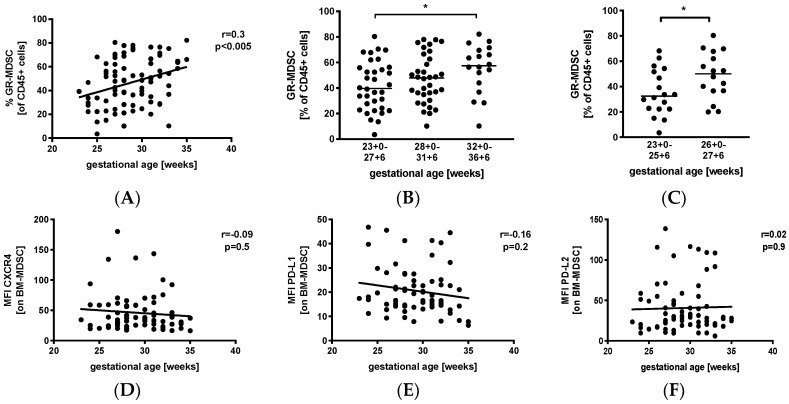
Quantification of granulocytic myeloid-derived suppressor cells (GR-MDSC) in the breast milk of preterm infants (dependency on gestational age). Milk cells were isolated from the breast milk of mothers of preterm infants during the first five weeks of life. Percentages of GR-MDSC (of total CD45^+^ leucocytes) and the expression of activation markers CXC-motif chemokine receptor 4 (CXCR4), programmed death ligand-1 (PD-L1), and programmed death ligand-2 (PD-L2) on breast milk containing a large population of GR-MDSC (BM-MDSC) were determined by flow cytometry. The mean percentages of BM-MDSC and the mean mean fluorescence intensity (MFI) of activation markers over the first five weeks were calculated. (**A**) Scatter diagram showing the percentage of GR-MDSC of total milk leucocytes depending on gestational age. Regression line shows the correlation between percentages of BM-MDSC and gestational age. *n* = 86, Spearman correlation. (**B**) Scatter diagram showing the percentage of GR-MDSC of milk leucocytes from mothers of preterm infants delivered at <28 0/7 WOG, between 28 0/7 and 31 6/7 WOG and >32 0/7 WOG. *n* = 18–34, * *p <* 0.05; Kruskal–Wallis test and Dunn’s multiple comparison test. (**C**) Scatter diagram showing the percentage of BM-MDSC from mothers of preterm infants delivered at <26 0/7 WOG, between 26 0/7 and 27 6/7 WOG, * *p* < 0.05; Mann–Whitney test. (**D**–**F**) Scatter diagrams showing the expression of CXCR4 (**D**), PD-L1 (**E**) and PD-L2 (**F**) on BM-MDSC depending on gestational age. Regression line shows the correlation between percentages of BM-MDSC and postnatal age. *n* = 86, *p* > 0.05, Spearman correlation.

**Figure 2 nutrients-12-02571-f002:**
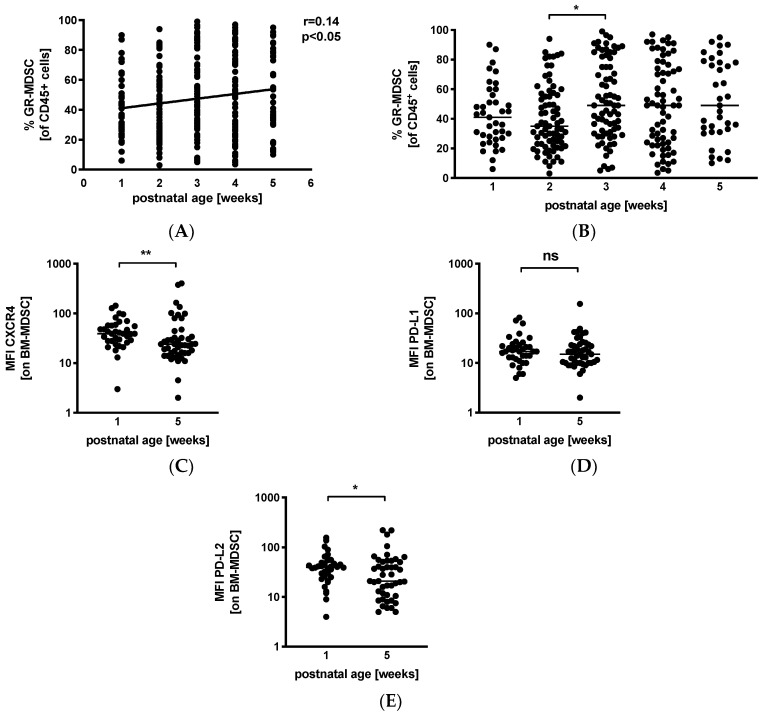
Quantification of GR-MDSC in breast milk of preterm infants (dependency on postnatal age). Milk cells were isolated from breast milk of mothers of preterm infants during the first five weeks of life. Percentages of GR-MDSC of total CD45+ leucocytes and the expression of activation markers CXCR4, PD-L1, and PD-L2 on BM-MDSC were determined by flow cytometry. (**A**) Scatter diagram showing the percentage of GR-MDSC from total milk leucocytes depending on postnatal age. Regression line shows the correlation between percentages of BM-MDSC and postnatal age. *n* = 288, * *p <* 0.05, Spearman correlation. (**B**) Scatter diagram showing the percentage of BM-MDSC in BM of mothers of preterm infants in postnatal weeks 1–5. *n* = 35–74, * *p* < 0.05; Kruskal–Wallis test and Dunn’s multiple comparison test. (**C**–**E**) Scatter diagrams showing the expression of CXCR4 (**C**), PD-L1 (**D**), and PD-L2 (**E**) on GR-MDSC in postnatal week 1 and week 5. *n* = 36–45, * *p* < 0.05, ** *p* < 0.005, ns not significant, Mann–Whitney test.

**Figure 3 nutrients-12-02571-f003:**
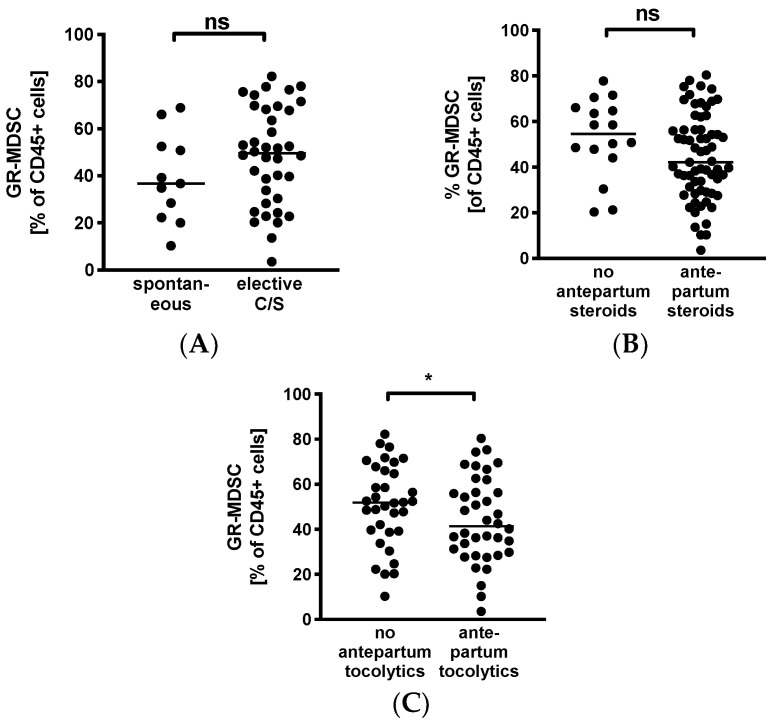
Quantification of GR-MDSC in the breast milk of preterm infants (maternal characteristics). Milk cells were isolated from the breast milk of mothers of preterm infants during the first five weeks of life. Percentages of GR-MDSC of total CD45^+^ leucocytes were determined by flow cytometry. The mean percentage of BM-MDSC was calculated. (**A**–**C**) Scatter diagrams showing the expression of GR-MDSC in breast milk of women who delivered spontaneously and of women who delivered by elective cesarean section (**A**), in women who received antepartum steroids and who did not (**B**), and in women who received antepartum tocolytics and who did not (**C**). *n* = 11–65, * *p* < 0.05, ns not significant, Mann–Whitney test.

**Figure 4 nutrients-12-02571-f004:**
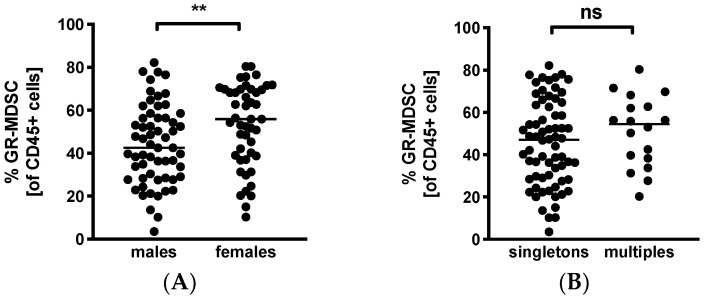
Quantification of GR-MDSC in breast milk of preterm infants (infant characteristics). Milk cells were isolated from the breast milk of mothers of preterm infants during the first five weeks of life. Percentages of GR-MDSC of total CD45+ leucocytes were determined by flow cytometry. The mean percentage of GR-MDSC was calculated. A+B Scatter diagrams showing the expression of GR-MDSC in breast milk of mothers of female and male infants (**A**) and singletons and multiples (**B**). *n* = 18–68, ** *p* < 0.005, ns not significant, Mann–Whitney test.

**Table 1 nutrients-12-02571-t001:** Infant characteristics of the study cohort.

	All (*n* = 105)	GA < 28 0/7 WOG (*n* = 45)	GA 28 0/7–31 6/7 WOG (*n* = 41)	GA > 32 0/7 WOG (*n* = 19)
GA (mean)	29	26 3/7	29 6/7	33 2/7
Birth weight (g) (mean)	1239	889	1327	1877
Gender male [%]	56.2	48.9	65.9	52.6
SGA (%)	6.7	2.2	12.2	5.3
Antenatal steroids (%)	77.1	88.9	75.6	52.6
Tocolytics (%)	46.7	60	36.6	36.8
MgSO_4_ (%)	32.1	46.7	22	21
Birth mode				
Spontaneous (%)	11.4	8.9	12.2	15.7
Elective C/S (%)	41	31.1	51,2	42.1
Emergency C/S (%)	47.6	60	36.6	42.1
Multiple pregnancies (%)	35.1	51.1	24.4	21.1
BPD (%)	10.5	22.2	2.4	0
NEC (%)	2.9	2.2	4.9	0
Sepsis total (%)	11.4	17.8	9.8	0
EOS (%)	3.8	2.4	7.3	0
LOS (%)	7.6	15.6	2.4	0

GA gestational age; SGA small for gestational age; C/S cesarean section; BPD bronchopulmonary dysplasia; NEC necrotizing enterocolitis; EOS early onset sepsis; LOS late onset sepsis.

**Table 2 nutrients-12-02571-t002:** Maternal characteristics of the study cohort.

	All (*n* = 86)	GA < 28 0/7 WOG (*n* = 34)	GA 28 0/7–31 6/7 WOG (*n* = 34)	GA > 32 0/7 WOG (*n* = 18)
GA (mean)	29 1/7	26 3/7	29 5/7	33 2/7
Antenatal steroids (%)	75.6	87.9	75	82.4
Tocolytics (%)	45.3	57.6	38.9	35.3
MgSO_4_ (%)	29.1	41.2	23.5	16.7
Birth mode				
Spontaneous (%)	12.8	12.1	11.1	17.6
Elective C/S (%)	44.2	36.4	52.8	41.2
Emergency C/S (%)	41.7	51.5	33.3	41.2
Multiple pregnancies (%)	20.9	33.3	13.9	11.8

GA gestational age; C/S cesarean section.

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
