# Peer review of "Granulocytic Myeloid-Derived Suppressor Cells in Breast Milk (BM-MDSC) Correlate with Gestational Age and Postnatal Age and Are Influenced by Infant’s Sex"

_nutrients, 2020, doi:10.3390/nu12092571_

Round 1

Reviewer 1 Report

This manuscript investigates the presence of granulocytic myeloid-derived supressor cells in breast milk (BM-MDSC). The study includes data collected over a bit more thab two years period from mothers and children born pre-term/term. BM-MDSC cells were quantified in milk during the first weeks of lactation as well as their activation state. The authors analysed the measurements according to different parameters such as gestational age, postnatal age, sex of the child, treatment administration, birth mode, etc. 

The findings of the authors conclude that BM-MDSC have a positive correlation with gestational age as well as postnatal age. However, activation markers on these cells did not change with gestational age and decreased with postnatal age. Furthermore, antepartum tocolytics and infant's sex had effects on BM-MDSC numbers. 

This findings will increase the knowledge on immune factors contained in breast milk, their role in the development of the immune function of the newborns and could help improving treatment strategies for premature newborns highly susceptible to infections. 

The content of the manuscript is correct, all sections include the required information and results are presented adequately. Nonetheless see few comments: 

Figure 1> graphs not aligned. Overlapping of graphs allows for seeing that graph a) has r and p values not readable. Sloppy figure. 

Figure 2> graphs wrongly numbered a, b, d, e f --> it should be a, b, c, d ,e

lines 190-192 > "This section may be divided..." --> I believe this sentence is example text and should be deleted

Funding section: "Please add: "This research received no external funding" or " this is example text and should be removed 

Reviewer 2 Report

To the authors,

This is a clinical study to evaluate the trends of granulocytic myeloid-derived suppressor cell counts in breast milk (BM-MDSC) and its correlation between gestational age, postnatal age, and infant’s gender. The study results may provide new information about the immune effect of breast milk to mitigate the threats of infection or sepsis in these vulnerable preterm infants.

Overall, this topic is interesting. However, several issues that should be mentioned clearly in the manuscript. The authors should discuss more about the possible mechanisms for the different MDSC in mothers with and without tocolysis. The different counts of MDSC by infant’s gender should also be discussed in detail. Besides, there were some incorrect presentation in the section of results should be revised.

I have some comments and suggestions for the authors and hope that my comments are constructive to the authors.

Introduction

  1. Line 53-59: please move the study findings in the section of the result.
  2. Please add some specific descriptions and more potential rationales for the differences and trends of MDSC in breast milk.

Method

  1. Line 64: It is confused here and easy to misunderstand that your study participants were mothers without severe complications or preterm infants without severe complications?

Results

  1. Line 111-112: “over the first five postnatal weeks” Did the authors compare the cell counts by the gestational age at birth? Or by the postmenstrual age? Please clarify it. 
  2. Line 113: the authors reported “We found that mean GR-MDSC counts positively correlated with gestational age at birth (p<0.005, r=-0.03, n=86, Figure 1A)” here. However, in Figure 1A, it showed the “% GR-MDSC” in the y-axis. Which one is correct?
  3. Line 144-145: why the authors reported the case numbers in “second to third week” together? Why not define the postnatal age as “the first postnatal week”, “second”, “third”, “fourth”, and “fifth” respectively?
  4. Line 147-148:“Separate analysis by postnatal week showed an increase in BM148 MDSC between week 2 (median 35.0%, n=74) and week 3 (median 49.0%, n=74, p<0.05).” The authors stated that the n numbers at postnatal week 2 and postnatal week 3 were 74. Is it true? Or n=74 at the postnatal 2nd to 3rd weeks? Please clarify it.
  1. Figure 2: please check the graphic orders and remarks. Where is the figure 2c?
  2. Line 168-176: Please add some rationales for the subgroup analysis in the introduction and discussion parts. The authors should provide more description for the reason to subgroup the study participants as “receive tocolytics or not” and “antepartum steroid or not”.
  3. Line 190-192: “This section may be divided by subheadings. It should provide a concise and precise description of the experimental results, their interpretation as well as the experimental conclusions that can be drawn.” Is this the comment from other reviewers or authors? Please remove it.

Discussion

  1. Please discuss more about the potential effects or mechanisms of antenatal steroid or tocolytic therapy on the level of MDSC in breast milk.
  2. I can’t understand the reason to choose infant gender in the subgroup analysis. Although there were some studies reported nutrients in breast milk may differ from neonates of different infant gender. Are there any immune mediators or biomarkers with similar findings as in your study? Please discuss more about the potential mechanisms and effects of infant gender on the breast milk production.

Round 2

Reviewer 2 Report

The authors have responded to my questions and comments satisfactorily. However, I have noticed some minor issues which should be revised or clarified.

Abstract
1. Line 20. The presentation of BM-MDSC was still inconsistent in the abstract and the main text. Should it be mean percentage of BM-MDSC or BM-MDSC numbers? Please revise it and check it in the manuscript.

Result

  1. Line 120. “r=-0.03” However, in the figure 1 A, it showed that r=0.03. Please check it.
  2. Line 137. “The mean of BM-MDSC numbers and ….” Should it be mean percentage of BM-MDSC? Please clarify or revise it.
  3. Line 153. “r=-0.14” However, in figure 2A, the correlation line seems to be a positive correlation. Please check it.

Discussion

  1. Line 270-275. The authors discussed about antenatal MgSO4 with the potential effect on reduced chemotaxis and motility of neutrophils. This is a good point for further study and discussion in the future. In your current study, for the convenience of readers, it would be better to add the information of antenatal MgSO4 therapy in the description of maternal characteristics (Table2).
